# How to improve patient safety in fragile, conflict-affected and vulnerable settings: a Delphi study protocol

Alexandra Shaw,[1] Niki O'Brien ![ORCID],[2] Kelsey Flott,[1] Sheila Leatherman,[3] Michael Durkin,[1] Ara Darzi,[1] Ana Luisa Neves ![ORCID][1,4]

¹Patient Safety Translational Research Centre, Imperial College London, London, UK
²Institute of Global Health Innovation, Imperial College London, London, UK
³Department of Health Policy and Management, UNC Gillings School of Global Public Health, University of North Carolina at Chapel Hill, Chapel Hill, North Carolina, USA
⁴Department of Community Medicine, Information and Health Decision Sciences (MEDCIDS) / Center for Health Technology and Services Research (CINTESIS), University of Porto, Porto, Portugal

**Correspondence to**
Ms Niki O'Brien;
n.obrien@imperial.ac.uk

## ABSTRACT

**Introduction**  There is a high burden of adverse events and poor outcomes in fragile, conflict-affected and vulnerable (FCV) settings. To improve outcomes, there is a need to better identify which interventions can improve patient safety in these settings, as well as to develop strategies to optimise their implementation.

**Objective**  This study intends to generate a consensus on the most relevant patient safety interventions from experts with experience on FCV settings, including frontline clinicians and managers/administrators, non-governmental organisations, policymakers and researchers.

**Methods and analysis**  The study uses an online Delphi research approach (eDelphi). Participants will include experts from a range of backgrounds, including those working in a variety of FCV settings. Participants will be established contacts known to the research team or recruited via snowball sampling, and will be asked to identify and rank the importance of a variety of patient safety interventions. Consensus will be defined as >70% of participants agreeing/strongly agreeing or disagreeing/strongly disagreeing with a statement. Data analysis will be completed in Microsoft Excel and NVivo. The primary outcome of the study will be a list of the most relevant and applicable patient safety interventions for FCV settings.

**Ethics and dissemination**  The study has received approval from Imperial College London Ethics Committee (reference number 20IC665). Anonymous results will be made available to the public, academic organisations and policymakers.

## Strengths and limitations of this study

► A key strength of this study is the utilisation of an eDelphi approach offering participation of geographically spread participants, and flexibility and anonymity to facilitate open discussion.

► This study has the potential to start building an evidence base and pave the way for the development and evaluation of safety interventions in these settings.

► This study has the potential to draw attention and funding to this area and contribute to policy and real-world interventions to improve the safety of care in fragile, conflict-affected and vulnerable settings.

► Limitations of the study include the potential exclusion of those working in the field due to unreliable or lack of internet access.

► We will attempt to mitigate higher attrition rates among participants, due to the nature of the work and conflicting priorities, with careful consideration of the participant list and by having as few rounds as possible, and a short survey length.

every year, it is estimated that 134 million adverse events and 2.5 million deaths occur globally.[1] While the need to improve patient safety is an increasingly recognised priority, there is still much to be done to develop and implement appropriate interventions to achieve this goal.[2]

The term fragile, conflict-affected and vulnerable (FCV) settings describes any setting of crisis, and may include armed conflict, complex emergencies, natural disasters and disruption to public services, and is recognised by the WHO and the World Bank.[3–5] The WHO publication *Quality of care in fragile, conflict-affected and vulnerable settings* (2020) defines FCV settings as settings 'experiencing humanitarian crises, protracted emergencies, prolonged disruption to critical public services or governance (eg, due to political or economic challenges, conflict or natural disaster), or armed conflict'.[4] Approximately 2 billion people are currently

## INTRODUCTION

The safety of patients is a critical element of well-functioning health systems globally—and an essential component to improve health outcomes and sustainably achieve universal health coverage (UHC). While the focus of health system strengthening has historically been on UHC, there has been a growing recognition that the implementation of interventions to improve quality and safety of care is equally critical to achieve better health outcomes. In low and middle-income countries, attention to patient safety is particularly important. In these settings, the prevalence of patient safety incidents is particularly high:

estimated to live in FCV settings with this figure set to rise.[6] Despite their heterogeneous nature, FCV settings share common challenges, including the 'disruption of routine health service organisation and delivery systems, increased health needs, complex resourcing landscapes, and vulnerability to further public health crises'.[4] These factors can pose, both individually and synergically, important threats to the safety of patients. Estimates suggest large numbers of preventable deaths take place in FCV settings, including 60% of preventable maternal deaths, 53% of deaths in children under 5 years and 45% of neonatal deaths.[4] Patient safety priorities and the facilitators and barriers to implementing patient safety interventions vary between and across individual FCV settings, reflecting the need for a flexible approach to improving safety and quality in service delivery.[4]

To reduce the impact of unsafe care on patient outcomes in FCV settings, patient safety interventions could be implemented to reduce incidents and improve the standard of care. Potential interventions cover a range of areas including, for example, infection prevention and control, health worker training and point of care. Incidents of unsafe care range from wrong site surgery to incorrect patient identification and delayed or missed opportunities for intervention resulting in preventable morbidity and mortality.[4]

An example of a specific intervention is the WHO Surgical Safety Checklist. In a study by Haynes *et al*,[7] the rate of death before and after the checklist was introduced declined from 1.5% to 0.8% and inpatient complications dropped from 11% to 7%.[7 8] Notably, the implementation of the surgical checklist requires a context-specific approach to maximise patient safety improvements.[9]

Despite the significant impact of patient safety incidents in FCV settings, there is limited evidence about the most applicable patient safety interventions, and how best to support these communities to improve the safety of care. Such a gap in knowledge makes it challenging to assess the quality and safety of existing interventions, measure their accessibility and effectiveness, address any context-specific challenges and develop new interventions.[4] In other healthcare settings it is possible to capture data and record it over time; however, in FCV settings this can be challenging due to system, local or organisational leadership instability, lack of technology and infrastructure and clinical priorities due to the demand for care on limited resources and staff. As such, a critical starting point must be to build an evidence base and identify which interventions are considered most useful and relevant in FCV settings—and to understand what is required to implement interventions that are both accessible and effective. It is unlikely that this evidence will initially come from traditional academic sources, and there is an opportunity to bring together and capitalise on the knowledge and experiences of experts working in these contexts and experts working on this topic area.

The aim of the study is to identify the most relevant patient safety interventions for these settings. The Delphi study design used by the research team will aim to answer the question 'Which interventions are needed to improve patient safety in FCV settings?', leveraging on the perspectives of frontline clinicians, managers/administrators, policymakers, researchers and non-governmental organisations (NGOs).

## METHODS AND ANALYSIS
### Study design
The study will use a Delphi methodology to seek consensus among a group of experts. Where there is a lack of research evidence and a desire to reach consensus, a commonly used formal consensus method is the Delphi technique, which uses rounds of questionnaires to collect data and achieve group consensus.[10] As the research will be conducted remotely online, through this paper we use the term 'Delphi' to describe the established research technique, and the term 'eDelphi' to describe our proposed research.

The eDelphi study will seek to gain consensus on the question 'Which interventions are needed to improve patient safety in FCV settings?' For the purposes of the study 'interventions' will be defined in the survey as 'the act of interfering with the outcome or course especially of a condition or process (as to prevent harm or improve functioning)'.[11]

### Survey development
Statements for the eDelphi (ie, brief descriptions of the interventions) will be developed from the study team's expertise, a rapid review of the literature and feedback from the eDelphi participants prior to the start of the Delphi process.

Rapid reviews have been recognised by the WHO as a useful approach to provide actionable, relevant and cost-effective evidence.[12] In rapid reviews, the steps of the systematic review are streamlined to produce evidence in a short time frame.[12] In a range of circumstances, including health policy research, there is value in accelerating knowledge synthesis for pressing policy and system decisions. The keywords used will be made as broad as possible to capture as many publications as possible on the topic of patient safety interventions in FCV settings (ie, 'patient safety' and 'intervention'). Search terms related to setting will be developed using the annually updated list of fragile and conflict-affected situations as defined by the World Bank Country and Lending Groups from 2011 to 2021.[13] Databases searched will include PubMed/Medline and Embase, as well as grey literature, particularly the outputs of NGOs. Papers published between 2011 and 2021, in English language, will be included.

The experts who will participate in the subsequent eDelphi will be asked to answer an exploratory question (free text) to list as many interventions as they want or a set number of responses to the question: 'Which interventions are needed to improve patient safety in FCV settings?' The questionnaire will be hosted on Qualtrics,

with a link to the survey distributed via email to all participants followed by two reminder emails, at 1-week intervals.

## Data collection

Qualtrics will be used to develop an online survey for each round of the eDelphi. The survey will be conducted across three rounds and each one will take approximately 15 min or less to complete. A link to each survey will be distributed via email to all participants followed by two reminder emails, at 1 week intervals, per survey round.

In round 1, participant demographics will also be collected, including: gender, year of birth, country of residence and current professional role. In rounds 1 and 2, experts will be asked to rank the interventions by answering the question: 'This intervention is relevant to improve patient safety in FCV settings'. Responses will be captured in a four-point Likert scale ('strongly agree', 'agree', 'disagree', 'strongly disagree'). The four-point Likert scale has been used in previous Delphi studies, producing stable findings.[14] For each statement, participants will be given the option to select 'don't know' as an alternative response. A free text response will also be made available to participants after each statement, facilitating them to elaborate on their thoughts or explain their responses. In round 3, experts will be presented with the individual and group results from round 2, to enable them to reconsider their responses in this round. With this information, participants will be asked to order each of the interventions that received consensus in rounds 1 and 2 based on which are likely to be most impactful. Responses will be subsequently analysed and ultimately mapped in thematic areas ranked by importance.

## Expert panel recruitment

Delphi studies employ a panel of expert members, as opposed to a random representative sample. In this study, 'expert' will be defined as a professional with experience working in health delivery and/or patient safety in FCV settings and experts working on this topic area. For the purposes of recruitment in the study, 'FCV settings' will be considered as per the definition of WHO (ie, settings experiencing humanitarian crises, protracted emergencies, prolonged disruption to critical public services or governance (eg, due to political or economic challenges, conflict or natural disaster) or armed conflict).[4] These settings can be entire countries (eg, Syria) where war has affected the national health system, or specific geographic areas of countries that have been affected (eg, the aftermath of a natural disaster, an influx of refugees to a particular area, etc). This definition enables the recruitment of experts across high, middle and low-income health systems to gather a broad range of opinions (see figure 1). We will recruit experts on the countries listed in the World Bank Fiscal Year 2022 (FY22) list of fragile and conflict-affected situations, as well as individuals working in these settings.[5] We will also recruit experts who currently work, or have previously worked, in settings at the intersection outlined

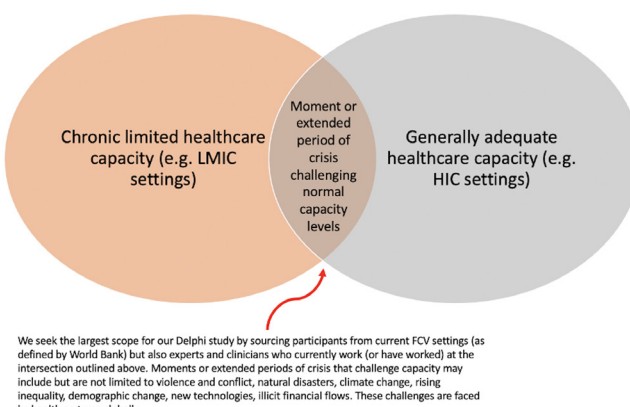

We seek the largest scope for our Delphi study by sourcing participants from current FCV settings (as defined by World Bank) but also experts and clinicians who currently work (or have worked) at the intersection outlined above. Moments or extended periods of crisis that challenge capacity may include but are not limited to violence and conflict, natural disasters, climate change, rising inequality, demographic change, new technologies, illicit financial flows. These challenges are faced by health systems globally.

**Figure 1** Conceptual framework for expert panel recruitment. FCV, fragile, conflict-affected and vulnerable; HIC, high-income country; LMIC, low and middle-income country.

in figure 1. Participants will be included if aged 18 years or older and able to understand English.

Purposive sampling will be used to ensure representativeness from a variety of expert roles (clinicians, managers/administrators, policymakers, researchers and NGOs) at both organisational and governmental levels. The majority of participants will be established contacts of the research team via the Institute of Global Health Innovation's Leading Health Systems Network and the National Institute for Health Research (NIHR) Imperial Patient Safety Translational Research Centre (PSTRC). Other relevant participants will be identified through the team's contacts at the WHO or using snowball sampling. Attention will be paid to ensure the range of participants is geographically diverse and reflects the wide range of FCV settings, including settings of current or recent armed conflict, complex emergencies, natural disasters and disruption to public services.

There is no agreement in the literature on the optimal number of participants to include in a Delphi survey.[15] Therefore, we will include all 'experts' who express an initial interest to participate. An initial invitation email will be sent to possible participants identified by the research team. Participants will be made aware that their involvement is voluntary, and if they do not take part in the study, it will not affect their relationship with the study team. Each participant will be given a Participant Information Sheet and provided with the opportunity to address any queries they may have with a researcher.

## Data analysis

Descriptive statistics will be used to describe participants' characteristics. The answers to the pre-Delphi question 'Which interventions are needed to improve patient safety in FCV settings?' (free text) will be analysed by two independent qualitative researchers using thematic analysis (NVivo software). To cross-check data analysis and ensure data quality, consistency in approach and transparency of analytical decision-making, 50% of the data will be read

and coded independently by a second experienced qualitative researcher.[16] Discrepancies in interpretation will be reviewed and resolved by a third qualitative researcher. A number of statements representing each theme will then be developed.

The answers to the Delphi rounds (ie, ranking safety interventions as outlined) will be described using mean, median, SD and IQR. Consensus will be defined as >70% of participants agreeing/strongly agreeing or disagreeing/strongly disagreeing with a statement. This level of agreement has been previously considered appropriate in similar Delphi studies.[17 18] 'Don't know' answers will be excluded from the analysis. The analysis will be completed using Microsoft Excel.

### Patient and public involvement

Patient partners will be included in the interpretation of our results, in the codevelopment of a dissemination strategy and in summarising the research findings into lay summaries and reports, in order to raise awareness and stimulate public participation on this topic. The research team at the NIHR Imperial PSTRC has a dedicated Patient and Public Involvement and Engagement (PPIE) lead and a Research Partners Group.

### Ethics and dissemination

This project has received ethics approval from the Imperial College London Ethics Committee (reference number 20IC665). Consent has been incorporated into the survey alongside the participant information provided. The results of this study will be published in a peer-reviewed journal and disseminated at national and international conferences.

## DISCUSSION

The aim of the study is to identify the most relevant patient safety interventions for FCV settings through an eDelphi research approach. Recognising the increasing scope of FCV settings, and their wide-ranging nature and impact on increasing numbers of the global population, we will identify critical actions to be taken forward.

### Strengths and limitations

There are both strengths and limitations for this study. The qualitative aspect of the study will enable the collection of detailed information from participants to answer the research question, and in addition may include the collation of relevant information that may not have been anticipated by the researchers.[19] It also offers the opportunity for the research team to explore participants' perceptions and the justifications for their responses in detail.[20] The use of a Delphi method specifically offers flexibility to adapt the research to reflect the unique and challenging topic, offers anonymity to the participants, and the opportunity to probe and develop existing knowledge and areas of controversy collectively.[21 22]

The nature of the eDelphi and the sampling strategy proposed will also enable the contribution of geographically diverse participants from regions around the world, and a good representation of those working in (and with lived experience) of these settings. The online nature of the eDelphi is further valuable in the context of the ongoing COVID-19 pandemic, as participation in the consensus building exercise will not be restricted by social distancing. Contributions will remain anonymous and will not be individually identifiable.

There are also several limitations to this work (see online supplemental appendix 1). Common critiques of the Delphi method tend to focus on the challenge of defining consensus, which can vary substantially from one Delphi research project to another, and the attrition rate of participants, which typically increases in each round of the study.[23] In conducting research in FCV settings, we would expect attrition rates to be particularly high due to the nature of work and conflicting priorities among participants. We will seek to mitigate this challenge by addressing the time commitment required from participants, including reducing the length of the surveys to the minimum. Similarly, in the first instance, we will seek a larger sample size to account for attrition.

The Delphi method has also been criticised for the role of response anonymity and the subsequent lack of participant accountability and ownership of ideas, which is further exacerbated in the eDelphi technique as physical anonymity is an additional characteristic.[21–23] Further, the fact that participants will not be in the same room to discuss their experiences and debate differences may lead to lack of engagement with the process. We will be mindful of this challenge and address any participant concerns through the course of the research. The nature of the eDelphi, specifically the use of online platform Qualtrics to administer the survey questionnaire instead of a traditional face-to-face approach, will require a stable internet source and internet access. Given the internet access and connectivity challenges in some of the most challenging FCV settings, some potential participants in the survey may be excluded from participating in the research. In the online supplemental appendix 1 document, we outline several mitigation strategies noted by Donohoe *et al* that may be relevant in the FCV content.[22]

Finally, the eDelphi has been criticised for the inherent loss of research control as the research is moved online, giving the research team less oversight on appropriate participation, including ensuring expert participation and adherence to protocols.[22] As highlighted (see online supplemental appendix 1), the development of a risk register may offer the research team a lens by which to assess the major threats to research control in the FCV setting including, but not limited to, FCV-based disruptions and security challenges.

### Implications for research and policy

This paper outlines the design of an eDelphi study to address the lack of research and evidence base for patient

safety interventions in FCV settings. The aim of the study is to seek an understanding and gain consensus on the interventions that may be most relevant in FCV settings. Recognising the increasing scope of FCV settings, this work aims to identify critical actions to be taken forward. This goal is aligned with the UHC agenda, which advocates for healthcare for all and prioritises the reduction of inequities in access to provision of care—and, ultimately, in safety outcomes.

The results of this study will create a list of the most relevant patient safety interventions, based on the consensus reached among a range of experts. The outcomes of this study have the potential to increase awareness in this area, and to identify interventions with a higher priority for implementation, as well as further evaluation and research.

**Contributors** ALN, AS and NO'B conceptualised the manuscript. ALN and AS planned the development of the work. ALN, AS and NO'B led the writing of the original draft manuscript, with input from KF, SL, MD and AD. All authors (AS, NO'B, KF, SL, MD, AD, ALN) equally contributed to the writing, reviewing and editing to develop the final draft. All authors approved the version submitted for publication.

**Funding** ALN is supported by the National Institute for Health Research (NIHR) Imperial Patient Safety Translation Research Centre (award reference: PSTRC_2016_004). Infrastructure support was provided by the NIHR Imperial Biomedical Research Centre (BRC).

**Disclaimer** The views of the authors do not necessarily reflect those of the NHS, NIHR or the Department of Health.

**Competing interests** None declared.

**Patient consent for publication** Not required.

**Provenance and peer review** Not commissioned; externally peer reviewed.

**ORCID iDs**
Niki O'Brien http://orcid.org/0000-0002-8389-1448
Ana Luisa Neves http://orcid.org/0000-0002-7107-7211

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
