## [Reviewer comments · BMJ Open]

ARTICLE DETAILS

TITLE (PROVISIONAL)	How to improve patient safety in fragile, conflict-affected, and vulnerable settings: a Delphi study protocol
AUTHORS	O'Brien, Niki; Shaw, Alexandra; Flott, Kelsey; Leatherman, Sheila; Durkin, Michael; Darzi, Ara; Neves, Ana Luisa

VERSION 1 – REVIEW

REVIEWER	Scott, Jason Northumbria University, Faculty of health and life sciences
REVIEW RETURNED	07-May-2021

GENERAL COMMENTS	This will be a valuable study of relevance to a generally under-represented area of global health. The use of an eDelphi approach (rather than Delphi) is entirely justified. The sampling strategy appears to take into account some of the challenges with conducting research in FCV settings, and I was really pleased to see the authors had addressed challenges of using the eDelphi approach along with their mitigation strategies, particularly the potential for digital exclusion. I have two minor suggestions for improving the manuscript:  1. In the introduction you give surgical checklists as an example intervention, and cite the seminal work of Haynes et al (2009). It may be beneficial to readers who are unfamiliar with surgical checklists and the surrounding research to also emphasise that implementation is often context-specific, which would further reinforce the need for your research in FCV settings. Weiser and Haynes (2018) recently wrote an excellent overview that encompasses this point. Weiser, T. G., & Haynes, A. B. (2018). Ten years of the surgical safety checklist. The British journal of surgery, 105(8), 927.  2. In the methods, framework analysis will be used for free text data. Please include a description of the framework that will be used in the deductive analysis.
--

REVIEWER	Haller, Guy Geneva University Hospital, Anaesthesia and Intensive Care/ Epidemiology
REVIEW RETURNED	31-May-2021

GENERAL COMMENTS	This protocol describes an expert consensus gaining study based on the Delphi methodology that aims at identifying relevant and applicable patient safety interventions that are the most important
---

	to the improvement of patient safety in fragile, conflict affected and vulnerable (FCV) settings. It also aims at gathering information on incentives and barriers for their implementation. While the protocol includes detailed information regarding study methodology and plans to include a broad range of experts, there are several issues that hinder the planned study to reach its goal. First, it is unclear what is embedded in the definition of “fragile, conflict affected and vulnerable (FCV) settings”. Are the authors describing areas of care in conflict zones (i.e. Syria, Afghanistan), or are they talking about healthcare systems in low income countries (i.e. Niger; Tchad) or about vulnerable population (i.e. disabled population; population with poor health literacy) ? This should be clearly defined since risk factors, interventions, barriers and enablers will significantly differ depending on the context and population included in the definition of FCV. Secondly, a Delphi study is not a qualitative study. It is a consensus gaining method based on a predefined set of questions that experts interviewed are asked to rate on a numerical scale. The protocol describes as the first round of the Delphi the following: st “experts will be asked to list as many interventions as they want or a set number of responses to the question: ‘What is needed in order to improve patient safety in FCV settings?’”. This is purely qualitative and not a formal Delphi process. While methodologically sound to develop the initial set of questions to be submitted to the experts for the Delphi process, this preliminary step of the study cannot be defined as "step 1" of the Delphi, It is not a Delphi but a preliminary qualitative study. I would here strongly advise to use also existing literature on the topic to elicit pertinent questions to be submitted to experts. Furthermore a qualitative theme interpretation grid should be prepared and more than 2 experts involved for the interpretation of experts interviews. Thirdly, the manuscript mentions also the use of a free text section based on the use of a SWOT-strength- weaknesses opportunities and threats methodology to help for the implementation of patient safety interventions. While useful, SWOTS methods are usually used as part of a nominal group technique and not in a Delphi. Once again the purpose of the Delphi is to gain consensus and not to elicit new ideas. Fourth, expert recruitment is also a key issue. If they are included in a Delphi they need to be not only “front line clinicians, managers/ administrators, NGOs, policy makers, and researchers” but also experts in the area of FCV. It is unclear how these experts will be identified. Another issue is the absence of a “statistical method” section to describe how experts’ answers rated on a 1 to 10 scale will be analysed and scores provided to experts for additional rating for round 3 and 4. Finally, an excellent project but at this stage with too many undefined goals and methodological weaknesses. I would recommend starting with a formal qualitative study to build up goals and definitions before launching an formal Delphi study.
--	--

VERSION 1 – AUTHOR RESPONSE

Reviewer 1 comments:

Reviewer: 1. In the introduction you give surgical checklists as an example intervention, and cite the seminal work of Haynes et al (2009). It may be beneficial to readers who are unfamiliar with surgical checklists and the surrounding research to also emphasise that implementation is often context-specific, which would further reinforce the need for your research in FCV settings. Weiser and Haynes (2018) recently wrote an excellent overview that encompasses this point.

Weiser, T. G., & Haynes, A. B. (2018). Ten years of the surgical safety checklist. *The British journal of surgery*, 105(8), 927.

Response: We agree with the reviewer that we need to emphasise that implementation, and subsequent improvements following interventions, is often context specific. As such, we have now used the Weiser & Haynes (2018) paper to illustrate that further research in FCV settings will be beneficial in determining the facilitators and barriers to improvements following the implementation of patient safety interventions in FCVs.

Reviewer: 2. In the methods, framework analysis will be used for free text data. Please include a description of the framework that will be used in the deductive analysis.

Response: As we note in more detail in response to Reviewer 2 below, we have removed the SWOT methodology element from the protocol. As a result, there is no longer the need to conduct deductive and inductive coding of free text responses and so text related to qualitative analysis has been removed from the manuscript.

Reviewer 2 comments:

Reviewer: First, it is unclear what is embedded in the definition of “fragile, conflict affected and vulnerable (FCV) settings”. Are the authors describing areas of care in conflict zones (i.e. Syria, Afghanistan), or are they talking about healthcare systems in low income countries (i.e. Niger; Chad) or about vulnerable population (i.e. disabled population; population with poor health literacy) ? This should be clearly defined since risk factors, interventions, barriers and enablers will significantly differ depending on the context and population included in the definition of FCV.

Response: In the introduction we now included the WHO (2020) definition of “FCV setting” and explain that the concept is recognised by both the WHO and other organisations, such as The World Bank. We have also updated the text in the introduction to specify that the particular context in different FCV settings will influence patient safety interventions and implementation. We see why the reviewer has made the point: the term FCV settings covers a range of heterogeneous settings, countries and health systems, where risk factors, interventions, barriers and enablers will vary. However, these settings do share important commonalities, as explained in the definition above, and have therefore been studied as a setting in previous research.

Reviewer: Secondly, a Delphi study is not a qualitative study. It is a consensus gaining method based on a predefined set of questions that experts interviewed are asked to rate on a numerical scale. The protocol describes as the first round of the Delphi the following: “experts will be asked to list as many interventions as they want or a set number of responses to the question: ‘What is needed in order to improve patient safety in FCV settings?’”. This is purely qualitative and not a formal Delphi process. While methodologically sound to develop the initial set of questions to be submitted to the experts for the Delphi process, this preliminary step of the study cannot be defined as “step 1” of the Delphi, It is not a Delphi but a preliminary qualitative study. I would here strongly advise to use also existing literature on the topic to elicit pertinent questions to be submitted to experts. Furthermore, a qualitative theme interpretation grid should be prepared and more than 2 experts involved for the interpretation of experts interviews.

Response: We agree with the reviewer that the previous Round 1 of the study where “experts will be asked to list as many interventions as they want or a set number of responses to the question: ‘What is needed in order to improve patient safety in FCV settings?’” is not part of a Delphi methodology. However, we believe exploring patient safety priorities with experts prior to the start of the Delphi will add value to the Delphi, and have therefore moved this section as part of the survey development. As such, we have restructured the Data collection and Analysis sections of the manuscript to explain that we will conduct preliminary exploratory work to determine possible patient safety priorities. This preliminary work will include a literature review and a short survey for experts to complete.

Following this work, we will begin a formal Delphi process with Rounds 1-2 asking experts to rank the thematic areas on a Likert scale (1-4). Participants will also be asked to determine the accuracy of the thematic areas of patient safety interventions outlined on a Likert scale (1-4). In Round 3, experts will be asked to order each of the interventions that received consensus in Rounds 1 and 2 based on which are likely to be most impactful. Responses will be subsequently analysed and ultimately mapped in thematic areas by importance.

Reviewer: Thirdly, the manuscript also mentions the use of a free text section based on the use of a SWOT-strength- weaknesses opportunities and threats methodology to help for the implementation of patient safety interventions. While useful, SWOTS methods are usually used as part of a nominal group technique and not in a Delphi. Once again the purpose of the Delphi is to gain consensus and not to elicit new ideas.

Response: As rightly pointed out by the reviewer, while useful, SWOTS methods are usually used as part of a nominal group technique and not in a Delphi. We agree with the reviewer that including a SWOT methodology is not part of a Delphi methodology and as such we have removed this element from the protocol.

Reviewer: Fourth, expert recruitment is also a key issue. If they are included in a Delphi they need to be not only “front line clinicians, managers/ administrators, NGOs, policy makers, and researchers” but also experts in the area of FCV. It is unclear how these experts will be identified.

Response: We see why the reviewer has made this point: front line clinicians, managers/administrators, NGOs, policy-makers, and research may not necessarily be experts in the area of FCVs. However, we intend to ensure a mix of topic experts across academia and healthcare organisations globally, as well as individuals working in healthcare institutions in FCV settings to ensure a holistic approach to gaining consensus on priority areas. As such we have updated the manuscript to reflect this. We now note “This study intends to generate a consensus on the most relevant and applicable patient safety interventions for these settings from experts on FCV settings, including: front line clinicians and managers/administrators, NGOs, policy makers, and researchers.”

In the methods section, under recruitment, we have outlined more clearly how we will target participants, utilising a figure to explain our thinking. We have expanded the paragraph to explain how recruiting participants from high-, middle- and low-income countries who have recently faced a moment or extended period of crisis challenging normal capacity levels, can aid the exploration of patient safety priorities in a range of FCV settings.

We also explain how we will capitalise on our previously established contact network (Institute of Global Health Innovation’s (IGHI) Leading Health Systems Network (LHSN), NIHR Imperial Patient Safety Translational Research Centre (PSTRC), World Health Organization) to recruit participants.

Reviewer: Another issue is the absence of a “statistical method” section to describe how experts’ answers rated on a 1 to 10 scale will be analysed and scores provided to experts for additional rating for round 3 and 4.

Response: We agree with the reviewer that the initial manuscript did not provide enough information on the analytical approach to the experts’ answers. We have added additional information explaining the approach during Rounds 1 and 2, where experts will be asked to rank the thematic areas on a Likert scale (1-4, without 0) and to determine the accuracy of the thematic areas of patient safety interventions. Data will be analysed using descriptive statistics, including means, medians, standard deviation (SD) and interquartile range (IQR), Consensus will be defined as > 70% of participants agreeing/strongly agreeing or disagreeing/strongly disagreeing with a statement. This level of agreement has been previously considered appropriate in similar Delphi studies. This approach has been used in previous studies, which we now reference in the edited version of the manuscript.

In Round 3, experts will be asked to order each of the interventions that received consensus in Rounds 1 and 2 based on which are likely to be most impactful. These responses will be analysed and ultimately mapped in thematic areas by importance.

Reviewer: Finally, an excellent project but at this stage with too many undefined goals and methodological weaknesses. I would recommend starting with a formal qualitative study to build up goals and definitions before launching a formal Delphi study.

Response: We are pleased that the reviewer considers this to be an excellent project as we believe that it is of great importance to some of the world’s most vulnerable communities. We also appreciate the reviewer’s comments on ensuring the research is as robust as possible, with clearly defined goals, and as such, have addressed the methodological weaknesses highlighted by both reviewers (as stated above). As specifically discussed above, given the nascent literature available in this area, we will be conducting preliminary exploratory research, including a literature review and survey exploring patient safety priorities, prior to the start of the Delphi will allow us to develop appropriate themes ahead of Round 1. With a clearer focus, we believe the formal Delphi study protocol we propose will add tremendous value to the body of research on patient safety in FCV settings.

VERSION 2 – REVIEW

REVIEWER	Scott, Jason Northumbria University, Faculty of health and Life Sciences
REVIEW RETURNED	04-Aug-2021
GENERAL COMMENTS	Thank you to the authors for making comprehensive changes to the manuscript and providing a detailed response. The manuscript is still lacking some structure around the methods that needs further work. The 'survey development' section would likely be improved by moving this section earlier, perhaps after the first study design paragraph. The second and third paragraphs could then go into a new 'data collection' section, which would come after sampling and before analysis. The last couple of sentences from the survey development section still refer to qualitative analysis. It would make more sense for this

	to be moved to the data analysis section (so that it begins with analysis relating to survey development). The final clarification needed is around the rapid literature review, which was not present in the original protocol. You need to provide detailed information on how this review will be conducted; is it scoping / systematic? What data sources will be used (primary empirical research, government documents etc) and over what time periods? How will these data sources be identified (databases, search engines)? You should provide sufficient information so that this review could be replicated.
--	---

REVIEWER	Haller, Guy Geneva University Hospital, Anaesthesia and Intensive Care/ Epidemiology
REVIEW RETURNED	09-Aug-2021

GENERAL COMMENTS	Authors have correctly answered to all points raised. Minor improvements could be made in the methodology section to clarify sentences such as "Statements for the survey were developed from the study team's expertise and a review of the literature . As part of the preliminary work, we will undertake a rapid literature review and ask the experts who will participate in the subsequent eDelphi to answer an exploratory question to list as many interventions as they want or a set number of responses to the question..... etc.." Due to a mix of past and present tense it is unclear what is part of the prospective study and what has already been done. The discussion section includes a table that would better stand as an appendix and its content briefly described in the discussion section. It is uncommon to provide table content details description in the discussion part of a manuscript. For other aspects, the manuscript has been nicely rewritten and formatted to include many methodological improvements and clarifications of definitions. I hope the planned study will provide interesting and useful findings in an area of need.
--

VERSION 2 – AUTHOR RESPONSE

Reviewer 1:

Comment 1. The 'survey development' section would likely be improved by moving this section earlier, perhaps after the first study design paragraph. The second and third paragraphs could then go into a new 'data collection' section, which would come after sampling and before analysis.

Answer: We agree with the reviewer that moving the “survey development” section earlier, as well as including a new “data collection” section would improve the flow of the methodology section of the paper. As such, we have restructured this section accordingly.

Comment 2. The last couple of sentences from the survey development section still refer to qualitative analysis. It would make more sense for this to be moved to the data analysis section (so that it begins with analysis relating to survey development).

Answer: We further agree with the reviewer that the final three sentences from the survey development section would benefit from being moved to the “data analysis” section and have edited the manuscript accordingly.

Comment 3. The final clarification needed is around the rapid literature review, which was not present in the original protocol. You need to provide detailed information on how this review will be conducted; is it scoping / systematic? What data sources will be used (primary empirical research, government documents etc) and over what time periods? How will these data sources be identified (databases, search engines)? You should provide sufficient information so that this review could be replicated.

Answer: The reviewer correctly points out that we have included a rapid literature review in the latest version of the protocol based on the previous comments from the reviewers. We agree that further information on the details of the rapid literature review should be provided. The edited version of the manuscript currently reads as follows:

“Statements for the eDelphi (i.e., brief descriptions of the interventions) will be developed from the study team’s expertise, a rapid review of the literature, and feedback from the eDelphi participants prior to the start of the Delphi process.

Rapid reviews have been recognised by the World Health Organisation as a useful approach to provide actionable, relevant, and cost-effective evidence [12]. In rapid reviews, the steps of the systematic review are streamlined to produce evidence in a short time frame [12]. In a range of circumstances, including health policy research, there is value in accelerating knowledge synthesis for pressing policy and systems decisions. The keywords used will be made as broad as possible to capture as many publications as possible on the topic of patient safety interventions in FCV settings (i.e. “patient safety” and “intervention”). Search terms related to setting will be developed using the annually updated List of Fragile and Conflict-affected Situations as defined by the World Bank Country and Lending Groups from 2011-2021 [13]. Databases searched will include PubMed/Medline and Embase, as well as grey literature, particularly the outputs of non-governmental organisations (NGOs). Papers published between 2011 and 2021, in English language, will be included.”

Reviewer 2:

Comment 1. Authors have correctly answered to all points raised. Minor improvements could be made in the methodology section to clarify sentences such as "Statements for the survey were developed from the study team’s expertise and a review of the literature. As part of the preliminary work, we will undertake a rapid literature review and ask the experts who will participate in the subsequent eDelphi to answer an exploratory question to list as many interventions as they want or a set number of responses to the question..... etc.." Due to a mix of past and present tense it is unclear what is part of the prospective study and what has already been done.

Answer: The reviewer correctly points out that there is a confusion of tenses in the “survey development” section around what is part of a prospective study and what has already been done; we have now revised this section and improved consistency. Additionally, as pointed out by reviewer 1, we provide additional details on the rapid review process. For further detail, please see answer to Reviewer 1, Comment 3.

Comment 2. The discussion section includes a table that would better stand as an appendix and its content briefly described in the discussion section. It is uncommon to provide table content details description in the discussion part of a manuscript.

Answer: We thank the reviewer for this useful comment. As suggested, we have removed the table from the document and is now present as an Appendix Supplementary File.

We also took this opportunity to streamline the narrative and improve consistency throughout the manuscript, after the changes suggested by the reviewers were implemented. We hope you will be satisfied with the updated manuscript.